# A Fast and Accurate Real-Time Vehicle Detection Method Using Deep Learning for Unconstrained Environments

**Annam Farid** [1,*], **Farhan Hussain** [1,*], **Khurram Khan** [2], **Mohsin Shahzad** [3], **Uzair Khan** [3] **and Zahid Mahmood** [3,*]

1   Department of Computer and Software Engineering, College of Electrical and Mechanical Engineering (CEME), National University of Sciences and Technology (NUST), Islamabad 44000, Pakistan
2   Faculty of Computer Science and Engineering, GIK Institute of Engineering Sciences and Technology, Topi 23460, Pakistan
3   Department of Electrical and Computer Engineering, COMSATS University Islamabad, Abbottabad Campus, Abbottabad 22060, Pakistan
*   Correspondence: annamfarid15@gmail.com (A.F.); farhan.hussain@ceme.nust.edu.pk (F.H.); zahid0987@cuiatd.edu.pk (Z.M.)

**Abstract:** Deep learning-based classification and detection algorithms have emerged as a powerful tool for vehicle detection in intelligent transportation systems. The limitations of the number of high-quality labeled training samples makes the single vehicle detection methods incapable of accomplishing acceptable accuracy in road vehicle detection. This paper presents detection and classification of vehicles on publicly available datasets by utilizing the YOLO-v5 architecture. This paper's findings utilize the concept of transfer learning through fine tuning the weights of the pre-trained YOLO-v5 architecture. To employ the concept of transfer learning, extensive data sets of images and videos of the congested traffic patterns were collected by the authors. These datasets were made more comprehensive by pointing various attributes, for instance high- and low-density traffic patterns, occlusions, and different weather circumstances. All of these gathered datasets were manually annotated. Ultimately, the improved YOLO-v5 structure becomes accustomed to any difficult traffic patterns. By fine-tuning the pre-trained network through our datasets, our proposed YOLO-v5 has exceeded several other traditional vehicle detection methods in terms of detection accuracy and execution time. Detailed simulations performed on the PKU, COCO, and DAWN datasets demonstrate the effectiveness of the proposed method in various challenging situations.

**Keywords:** machine learning; object detection; vehicle detection

## 1. Introduction

The Human Vision System (HVS) reliably and accurately performs complex tasks, such as being able to detect and recognize and identify diverse range of objects with little conscious attention. With the recent developments in the Computer Vision (CV) and Machine Learning (ML), and with the availability of capabilities, such as massive data sets, faster GPUs, and better algorithms, it has now become possible for computers to detect, recognize, and classify several items in an image or video with high accuracy [1]. The aim of vehicle detection and classification is to locate vehicles in either images or videos [2]. Efficiency of vehicle localization is a critical step in traffic monitoring or surveillance. Figure 1 shows several detected vehicles from Pakistani traffic images that are achieved using the machine learning algorithms. Therefore, autonomous vehicle detection methods must exactly detect traffic objects, such as cars, vehicles, or police vans or bikes in real-time to gain good control and make right decisions for the public safety [3].

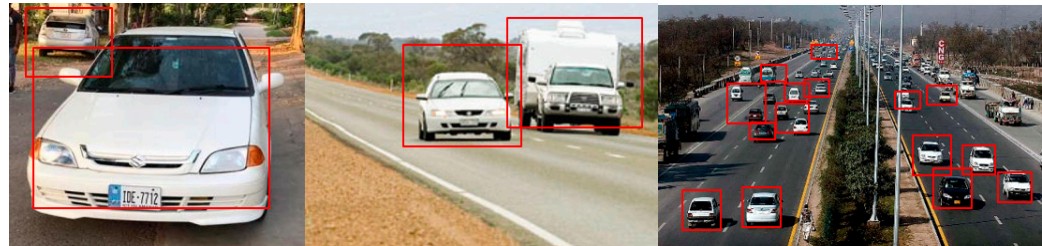

**Figure 1.** Vehicle detection examples from Pakistani traffic.

With the development of the DNNs, automatic vehicle detection has made substantial progress in recent years, for instance, in Autonomous Driving Systems (ADS) and driver support systems in the context of concerns about traffic congestion and driving safety [4].

To develop intelligent and autonomous systems, for instance, self-directed driving, surveillance, detecting objects, or tracking, vehicle localization is a crucial problem [5]. Automatic driving is a new high technology invention that relies on the ability to only find vehicles [6]. In the metropolitan areas, frequent incidents happen regarding traffic breaches, vehicle mishaps, and thefts that are recorded through the CCTV cameras. Traffic surveillance system detector should be fast, accurate, and reliable enough to detect vehicles in real-time. In the areas of traffic managing systems or surveillance technology, there have been numerous advancements. Two essential conditions are normally considered to rate vehicle detectors, which are its real-time detection ability and whether it has a high detection accuracy of the traffic objects under adverse weather conditions.

One of the interesting efforts to locate vehicles is to detect abnormalities in traffic violations, as well as careless driving on the roads. With the introduction of new technologies in the ITS and the growing demand for automation, the employment of technology in a variety of disciplines has become inevitable. Because of the growing number of cars on the road, automated vehicle traffic monitoring is one of the most important applications being developed for speed or traffic control, offence detection, road tolls, and a variety of other related issues. To manage such issues, large amounts of general budget is consumed. In large and congested cities, traffic surveillance is a major challenge. ITS mobility planning traffic engineering applications have made significant progress in reducing city incidents. Surveillance systems nowadays use traffic flow data, which typically consider crucial factors, such as speed, size, trajectory, and vehicle type. Moreover, vision-based systems are also used nowadays to monitor and record various traffic patterns.

Due to the developments in the DNNs, the ML based models can be reliably used to detect various vehicles, although the training speed of deep learning networks is much slower in the CPU calculations. However, the training time is significantly much less thanks to developing technologies, such as GPUs and the TPUs. When compared to standard ML-based approaches, the DNNs have significantly enhanced performance in various scenarios, such as smart self-governing, self-driving vehicles, intelligent observations, and smart city-based applications. DNNs that are based on neural networks constitute an advanced category of machine learning, which is very handy at resolving difficulties in a variety of complex models that usually are hard to explain through typical statistical techniques.

Moreover, the CNN, which is a form of deep neural network, is extensively used for image recognition and categorization. These are the algorithms that can identify various objects, such as, license plates, cars, people, and a variety of other objects [5]. A primary benefit of the CNN is that it extracts essential features without any human interaction after the training process. Different versions of the CNN, such as R-CNN, Fast-RCNN, and Faster-RCNN are the most popular and commonly utilized CNN approaches [5]. However, the computational load is still too high for devices with limited computing power and space to process photos. The D-based algorithms have long been regarded as effective tools for image recognition. The CNN-based methods have been frequently used in recent approaches among the many vehicle detection algorithms and are divided into region-based and regression-based methods.

The YOLO is a new method to detect diverse vehicles in a single step. The YOLO handles the vehicle perception problem as a regression problem by classifying the image via CNN, which is utilized to achieve robust vehicle detection. The YOLO can retrieve the object's position, category, and confidence score, as well as boost detection speed and detect the motion-blurred vehicles in real-time. A regression-based YOLO technique is one of the most current ways to predict bounding boxes and class probabilities directly in a single neural network. As a result, the YOLO model was created to speed up the process to identify an object and find its location in an image. It uses the CNN to detect several items in an image immediately. To handle vehicles of different shapes and sizes, it integrates predictions from many feature maps with different resolutions. With advancements in the YOLO based methods, such as YOLOv3 and YOLOv5, the YOLO continues to provide higher performance in terms of processing time and accuracy [3].

In this work, we target the detection and classification of vehicles in images using deep learning to explore the feasibility of YOLO based methods. The YOLO family of algorithms is first-order object detection method, which uses an anchor box to integrate various objects localization. Up to now, five versions of YOLO family of algorithms have been released. The YOLOv3 is a milestone in the performance and speed of the YOLO family of algorithms. Our motivation to choose the YOLOv5 detection model is due to its smaller architecture and much fast detection ability than the previous generations of its model families. Recently, researchers in various research domains have enhanced the original YOLOv5 model based on the characteristics of their detection targets, which makes the YOLOv5 algorithm an excellent choice in vehicle detection domain. Our main contributions in this work are listed below.

- We propose a modified version of the YOLO algorithm to achieve vehicle detection in real time. Earlier-developed works have been trained on massive datasets, but still need to be fine-tuned for use in congested traffic environments. However, we augment these datasets with our gathered datasets. We compare the efficiency of our trained version with several recent state-of-the-art methods.
- We detect and classify vehicles in images that are captured in various traffic scenes. We perform detailed study on the PKU, COCO, and DAWN datasets. To achieve higher accuracy on images from our local traffic patterns, we gathered an extensive dataset and applied transfer learning to the YOLOv5. The input to a system is a real-time image, and the output is a bounding box corresponding to all objects in the image, along with the class of object in each box.
- In addition, we employ a transfer learning approach to utilize the knowledge embedded in our local datasets. We believe that the ITS based applications require rapid and precise vehicle identification and classification. It is a challenging task to detect different vehicles abruptly and precisely due to short gaps between vehicles on the road and interference aspects of pictures or video frames containing vehicle images. Therefore, we are optimistic that our developed method provides a good insight into locating vehicles in congested traffic environments.

This paper is organized as follows. Section 2 discusses few recent related works. Section 3 describes in detail the proposed method. Simulation results and discussions are presented in Section 4. Finally, Section 5 concludes the paper and hints towards future research directions. For readers' smooth understanding, Table 1 lists the nomenclature that is used extensively in this paper.

**Table 1.** Nomenclature.

| Acronym | Meaning |
| --- | --- |
| CNN | Convolutional Neural Networks |
| COCO | Common Objects in Context |
| DLT | Dark Label Tool |
| DNN | Deep Neural Network |
| FPN | Feature Pyramid Network |
| FPS | Frames Per Seconds |
| HDT/LDT | High Density Traffic/Low Density Traffic |
| ITS | Intelligent Transportation Systems |
| LIT | Label Image Tool |
| mAP | mean Average Precision |
| MSR | Multi Scale Retinex |
| PAN | Path Aggregation Network |
| PKU | Peking University |
| R-CNN | Region-based Convolutional Neural Networks |
| RFW | RoboFlow |
| SSD | Single Stage Detector |
| TP/TN | True Positives/True Negatives |
| XAI | Explainable Artificial Intelligence |
| YOLO | You Only Look Once |

## 2. Related Work

Vehicle detection has gained considerable attention in the research community in the past two decades. In this section, we briefly discuss the recent advances in the vehicle detection domain. For readers' fair understanding, we categorize the literature into two streams as illustrated below.

### 2.1. Conventional Methods

This section quickly lists a few of the latest conventional vehicle detection approaches. In [6], the developed method detects vehicles in airborne images. In this work, the vehicle localization is attained through the Gaussian Mixture Model (GMM) and background subtraction representations. In [7], an ensemble-based method is developed for various image descriptors, which illustrate the distributions of gradients, color models, and textures. This work reports good results in high resolution aerial images. In [8], a new methodology through the application of the GMM is developed to detect dissimilar complex structures, for example, objects in residential, agricultural, and industrial zones. This work also reflects spectral and spatial constraints. An efficient, GMM-based image segmentation method is utilized in [9]. This method is capable of detecting the frontal view of different vehicles. To locate the vehicles' driving area, lanes are spotted through the application of the Canny edge detector along with Hough transform. To further enhance the efficiency of proposed method, this work uses the HOG features, colors, and the Harr-features of vehicles, and trains the SVM classifier. In [10], the SVM is trained through multi-feature fusion that results in reduced vehicle detection time. In [11], vehicle detection is achieved through integration of the SIFT with the SVM. To further improve classification ability, an integration of pyramids pooling, sliding windows, and NMS is done that substantially enhances the vehicle detection outputs, which are obtained therein.

### 2.2. YOLO-Based Methods

In [12] a vision-based object detection and recognition framework for autonomous driving was proposed with particular emphasis on: (i) an optimized model based on the structure of YOLOv4 was presented to detect 10 types of objects; (ii) a fine-tuned part affinity fields approach was developed; (iii) eXplainable Artificial Intelligence (XAI) was integrated to assist the approximations in the risk evaluation phase; (iv) an intricate self-

driving dataset was developed, which included several different subsets for each relevant task; and (v) an end-to-end system with a high-accuracy model was discussed.

The overall parameters of enhanced YOLOv4 are reduced by 74%, which meets the real-time capacity. Moreover, when evaluated with other methods, the detection precision of the enhanced YOLOv4 improved by 2.6%. In [13], a novel and efficient detector named YOLO-ACN is developed, which is inspired by the high detection accuracy and speed of YOLOv3. This technique is improved by the addition of an attention mechanism, a CIoU (complete intersection over union) loss function, Soft-NMS, and depth wise separable convolution. In this method, initially, the attention mechanism is built in the channel and spatial dimensions in each residual block focus on small targets. Later, CIoU loss is adopted to achieve accurate bounding box regression. Besides, to filter out a more accurate BBox and avoid deleting occluded objects in dense images, the CIoU is applied in the Soft-NMS, and the Gaussian model in the Soft-NMS is employed to suppress the surrounding BBox. Finally, to improve the detection speed, standard convolution is replaced by depth wise separable convolution. Meanwhile, a hard-swish activation function is utilized in deeper layers.

In [14], a multi-stage object detection architecture, which authors refer as Cascade R-CNN, is developed to address objects appearance and detection. The proposed R-CNN is composed of a sequence of detectors that are trained with varying IoU thresholds, to be sequentially more discriminating against close false positives. These detectors are trained stage-to-stage and by leveraging the scrutiny that the output of a detector is a good distribution for training the next higher stage detector. The resampling of improved hypotheses assures that all detectors have a positive set of examples of equivalent size, and thus reducing the overfitting. The same systematic method is applied at inference, enabling a closer match between the hypotheses and the detector quality of each stage. A simple implementation of the Cascade R-CNN is shown to surpass all single-model object detectors on the challenging COCO dataset. Simulations also reveal that the Cascade R-CNN is widely applicable across detector architectures and achieves consistent gains of the baseline detector strength.

A method to detect smoky vehicles with high precision and speed has been proposed in [15] using an enhanced lightweight network based on Yolov5. This work uses Mobilenetv3-small modified Yolov5s' backbone to reduce the number of model parameters and calculations. A vehicle exhaust dataset is collected and created to detect motor vehicle exhaust with high precision. Cutout and saturation transformations were used to enlarge the self-built dataset, which was eventually expanded to 6102 photos, due to the interference of vehicle shadows and occlusion between vehicles. The results demonstrate that applying data augmentation improves detection accuracy by 8.5%. The upgraded network is installed on embedded devices and has a detection speed of 12.5 FPS, which is two times faster than Yolov5. Only 0.48 million network parameters have been improved. This study suggests an effective target detection model as well as a strategy for developing low-cost and quick vehicle exhaust detection equipment. An effective nighttime vehicle detection approach is developed in [16]. First, an optimal MSR algorithm was used to improve the original nighttime photos. The improved photos were then used to fine-tune a pre-trained YOLO v3 network. Finally, the network was employed to distinguish vehicles from each other and outperforming two popular object detection approaches, the Faster R-CNN and SSD, in terms of precision and detection efficiency. The suggested method has an average precision of 93.66%, which is 6.14% and 3.21% higher than the Faster R-CNN and SSD, respectively.

In [17], the proposed work contributed to the field of autonomous driving through the DL techniques to detect objects. This work primarily uses the YOLO to locate numerous objects on the roads and categorized into the type that they belong to with the aid of bounding boxes. The YOLOv4 weights are used to custom train the model to detect the objects, and the data is acquired using the OIDv4 toolkit from the open-source data collection. In [18], an updated YOLOv3 algorithm for vehicle detection is developed. Initially, it clusters the data set using a clustering analysis approach, then optimizes the

network structure to raise the number of final output grids and boost the comparatively low vehicle prediction ability. It also optimizes the data set as well as the input image. Its robustness under various external situations is due to its resolution. Experiments demonstrate that the modified YOLOv3 algorithm outperforms the traditional approach in terms of detection accuracy and rate. In [19], researchers proposed the newest YOLOv3 algorithm to detect traffic participants. They trained the network for five different object classes, which are vehicle, truck, pedestrian, traffic signs, and lights. This work also discusses the range of driving scenarios that include bright and overcast sky, snow, fog, and night conditions. In [20], the baseline YOLO is used to detect moving cars. Meanwhile, a modified Kalman filter method is used to dynamically track the detected vehicles, which results in overall competitive performance in both day and night. The testing results reveal that the system is resistant to occluding vehicles or congested highways, with an average vehicle counting accuracy of 92.11% at the rate of 2.55 FPS. In [21], researchers suggested an updated Yolov3 transfer learning-based deep learning algorithm for object detection. In this work, the network is trained on a difficult data set, and the output is fast and precise, which is beneficial for applications that need object detection. In [22], a method is proposed that classifies vehicular traffic on video using a neural network. The necessity to regulate traffic on the roads has emerged as the number of vehicles on the road has increased, resulting in traffic congestion and a high accident rate. Collecting data from video of vehicles on the road will aid in the creation of statistics that can be used to efficiently consider traffic regulation on the roads. The challenge of vehicle categorization on video was solved using the YOLOv5 powerful real-time object classification method. For neural network training 750 images from outdoor surveillance camera were used as a dataset. After testing the model, the recognition accuracy was 89%.

YOLOv2 and YOLO9000 models were discussed in [23]. Their strength in real-time detection and classification of objects in videos made them useful in several applications. The YOLOv2 is very efficient at detecting and classifying simple objects. The GPU features and the Anchor Box approach were used to accomplish the desired speed and precision. Furthermore, YOLOv2 can accurately detect object movement in video recordings. YOLO9000 is a real-time framework that can maximize detection and classification while also bridging the gap. The YOLOv2 model and the YOLO 9000 detection system can detect and classify a wide range of items, from multiple occurrences of a single object to multiple instances of various objects. In [24], an improved YOLOv4-based video stream vehicle target detection system was used to address the problem of slow detection speed. This study first presents a theoretical overview of the YOLOv4 algorithm, then offers an algorithmic technique for increasing detection speed, and lastly conducts real road tests. According to the experimental results, the algorithm in this work can improve detection speed without sacrificing accuracy, which can be used to make decisions for safe vehicle driving.

In [25], the YOLOv5 is used to locate weighty supplies vehicles during cold weather and thus allowed the prediction of parking place slots in real-time. The authors employ infrared network cameras, since snowy conditions and the polar night in the winter pose certain obstacles for image recognition. Authors used the YOLOv5 to analyze if the front cabin and back are adequate features to identify heavy goods vehicles because these photos repeatedly have large overlaps. The trained algorithm reliably distinguishes the front of heavy goods vehicles. However, detecting the back cabin appears to be more difficult, especially when the vehicle is placed far away from the camera. Finally, they show that detecting heavy goods vehicles utilizing their front and rear instead of the entire vehicle improves detection during winters, which mostly experience difficult images with significant objects overlaps and cut-offs.

Recently, some of the learning-based approaches [26] and the CNN based methods [27] also report encouraging results in the vehicle detection domain. In [26], authors developed a box-free instance segmentation method using semi-supervised iterative learning. The iterative learning procedure considered labeling vehicles from the entire scene and then trained the deep learning model for classification. Authors also considered vehicle inte-

riors and borders to isolate instances using a semantic segmentation. In [27], researchers performed a fully convolutional regression network. In this method the training stage uses an input image along with its ground to describe each vehicle as a 2-D Gaussian function distribution. Hence, the vehicle's original format attains a simplified elliptical shape in the ground truth and output images. The vehicle segmentation uses a fixed threshold in the predicted density map to generate a binary mask. This method prevents grouping cars and favors counting. Moreover, vehicles take on a different form that is expressed by the Gaussian function.

In [28], a robust vehicle detection model is developed, which is referred to as YOLOv4_AF. This model introduces an attention mechanism that suppresses the interference features of images through channel length and spatial dimension. In addition, a modification of the Feature Pyramid Network (FPN) part of the Path Aggregation Network (PAN) is also applied to enhance the effective features. This way, the objects are steadily positioned in the 3D space that ultimately improves the vehicle object detection and classification performance. In [29], vehicle detection and tracking are achieved through a multi-scale deep convolution neural network. This work also applies conventional Gaussian mixture probability hypothesis along with hierarchical data association that divides into detection-to-track and track-to-track associations. Moreover, the cost matrix of each stage is resolved using the Hungarian algorithm. Only detection information is used in the previous so as to achieve rapid execution. In [30], Faster-RCNN is tuned to detect vehicles in various scenarios. Moreover, this work also uses basic image processing methods along with morphological operations and multiple thresholding to achieve vehicle exact location in near-real-time. In [31], vehicle and distance detection method is developed in a virtual environment. This work mainly uses the Yolo v5s neural network structure and develops a novel neural network system, which the authors refer as the Yolo v5-Ghost. In the discussed approach, the authors further fine-tuned the network layer structure of the Yolo v5s. Experiments performed therein indicate that this method is suitable to be deployed in real-time environments. The authors of this work also claim that their work is suitable for embedded and edge devices and object detection in general [32]. In [33], a novel bounding box regression loss approach is developed that learns objects bounding box through miscellaneous transformations and variance localizations. The learned localization variance is further merged during non-maximum suppression that increases the localization performance. In [34], a dynamic vehicle detection method, which is based on a likelihood-field-based model and on Coherent Point Drift (CPD), is developed. This study also applies an adaptive thresholding on the distance and grid angular resolutions to detect the moving vehicles. This work also presents the pose estimation that is based on the CPD to estimate the vehicle pose. The scaling series algorithm is also coupled with a Bayesian filter to update the vehicle localization states during various intervals.

In [35], a new Multi-Level Feature Pyramid Network (MLFPN) is proposed that constructs effective feature pyramids to detect objects. This method initially fuses multi-level features and later feeds the base features into a block of alternating joint thinned U-shape networks. Meanwhile, the decoder layers are gathered up with correspondent sizes to build a feature pyramid for object detection. In [36], the proposed method is primarily based on Trident Network (TridentNet), which aims to generate scale-specific feature maps. This scheme also constructs parallel multi-branches in which each branch shares the same transformation parameters. This algorithm also adopts a scale-aware training scheme to specialize each branch by sampling object instances of proper scales for training. The proposed TridentNet achieves significant improvements without any additional parameters. In [37], a single-stage method uses Mask SSD to investigate objects. This work uses a convolutional series to predict pixelwise objects' separation. This work also optimizes the whole network through multitask loss function. Ultimately, the network directly predicts final objects presence results. This work also uses multi-scale and feedback features that perform well on various objects of different scales and aspect ratios. In [38], the developed method uses two classifiers to tackle the problem of failure to locate vehicles

that have occlusions or slight interference. It accomplishes vehicle detection through a local binary pattern along with a support vector machine. This method also uses the CNN in the second phase to remove the interference areas between vehicles and any moving object.

In [39], a novel CornerNet is developed achieve accurate object detection. The CornerNet approach detects objects bounding box as a pair of keypoints. The top-left corner and the bottom-right corners are localized through a single CNN. Through an intelligent paired keypoints approach, this method eliminates the need to design a set of anchor boxes that are normally used in prior single-stage detectors. This work also introduces corner pooling, which is a new type of pooling layer and helps the network to better visualize and localize the objects' corners. In [40], a novel approach, which authors refer to as Mask R-CNN, is discussed that extends Faster R-CNN by adding a new branch to predict an object mask. The Mask R-CNN is simple to train and adds only a small overhead to Faster R-CNN. Moreover, Mask R-CNN is easy to generalize to other tasks, for instance to estimate object orientation in the same framework. This method is conceptually simple and flexible and efficiently detects objects in an image. In [41], an anchor-free vehicle detection approach is developed that is capable of detecting arbitrarily oriented vehicles in high-resolution images. This work considers vehicles as a multitask learning problem and predicts high-level vehicle features via a fully convolutional network. In this work, initially, coarse and fine feature maps outputted from different stages of a residual network are integrated through a feature pyramid fusion. Later, four convolutional layers are added to predict possible vehicle features. In [42], a scale-insensitive CNN (SINet) is proposed to locate vehicles with a large variance of scales. Initially, a context-aware RoI pooling is done to maintain the contextual information and original structure of objects. Later, a multi-branch decision mechanism is introduced to minimize the intra-class distance of various features. The proposed techniques can be further equipped with any deep network architectures and keep them trained end-to-end.

The preceding discussion offers a good suggestion that vehicle detection is a crucial step to develop systematic mechanisms, such as an intelligent transportation system. The methods describe above are a few of the efficient and good works that aim to address the vehicle detection problem in various environments. As we will see in Section 4, different datasets are publicly available to address the vehicle detection problem under diverse conditions. We believe that our work is an efficient addition in the vehicle detection domain. In the next section, we discuss our developed vehicle detection method.

## 3. Proposed Method

In this section, we describe our proposed method in detail. As discussed below, we divide our developed method into the following interconnected steps along with a brief description. Figure 2 shows the flow of the proposed method. In addition, Algorithm 1 shows more details of our developed method.

To test our method, we gather our own dataset from challenging Pakistani traffic environments. This dataset was collected over a period of two months in different cities of Pakistan. As shown in Figure 2 that the gathered data is preprocessed and augmented. Later it is trained by our model. Meanwhile, the YOLO-v5 model is built and trained. Our collected data is from an unknown distribution in Pakistani traffic. Therefore, it is now tested on the YOLO-v5 model. After the YOLO-v5 is applied, we then investigate and analyse our detector. To aid readers' understanding, below we describe the steps and details of our developed method.

---

**Algorithm 1:** Pseudo code of the proposed vehicle detection algorithm

---

1.  **Input:** A test image with one or more visible vehicles.
2.  Execute the algorithm in following order to get the desired result.
3.  **begin**
4.      **Gather data**
5.      **do**
6.          Categorize data into LDT and HDT scene images as indicated by Figure 3a,b.
                                              ► use the LIT
7.          Annotate data to identify classes of objects as shown by Figure 4a,b.
                                     ► use the DLT for video dataset
8.      **end**
9.    **begin** data preprocessing and augmentation
10.     **do processing**
11.      Preprocess that data and split into train, validation, and test set
                                         ► use the RFW tool
12.     **do augmentation**
13.      Standardize the image and video data from step (3) to step (8) up to 416 × 416 pixels.
14.      Crop dataset between 0% and 30% zoom.
15.      Saturate dataset between ±25%.
16.      Vary brightness, such as darken and brighten the images between ±25%.
17.       **end**
18.   **end**
19.   **begin** YOLO-v5
20.       **do**
21.     Install all Yolov5 repositories to be ready for running object detection training & inference.
22.     Download custom Yolov5 Object detection data.
23.     Configure model and architecture.
24.      **begin** Training
25.       Train custom YOLO-v5 detector           ► use YOLO-v5 architecture
26.       Use training parameters as:            ► use COCO dataset weights
27.        image size: 416 × 416 pixels,
28.        batch size: vary as 5, 10, and 20,
29.        epochs: vary as 100, 300, and 500,
30.        Configuration: use as per YOLO-v5s, YOLO-v5m, or YOLO-v5L,
31.        Weights: use pre-trained COCO dataset,
32.       **end**
33.     Run YOLO-v5 inference on test images.
34.       python detect.py --weights runs/train/exp/weights/best.pt,img 416,conf 0.1.
35.     **end**
36. **end**
37. **Output:** An image with detected vehicles through a bounding box around.

---

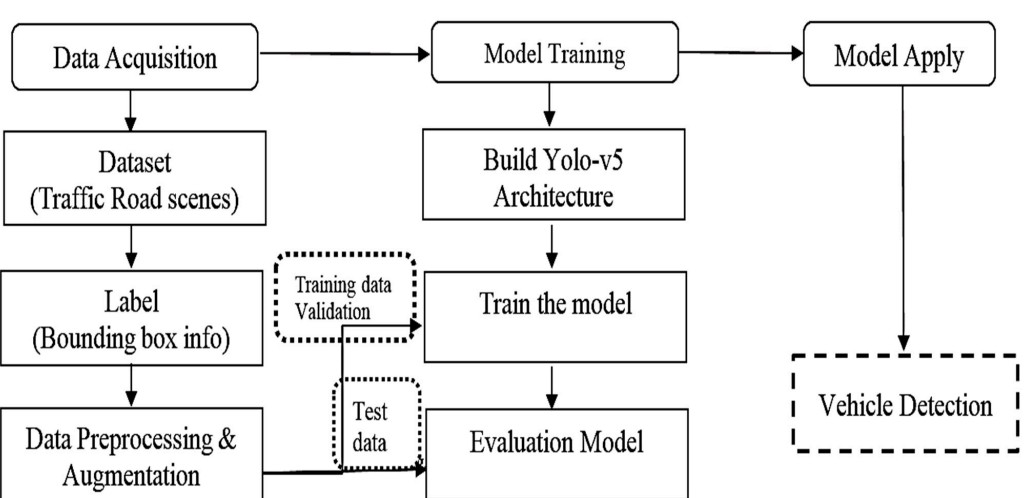

**Figure 2.** Flow of the proposed method.

### 3.1. Data Acquisition

To begin with the proposed algorithm, we initially acquire data. First we deal with different conditions on highways. For example, we come across the multi-class objects, such as different types of vehicles, motor bikes, and pedestrians on the roads. Similarly, we also faced severe and crucial challenges, such as massive traffic jams and overlapped vehicles. Therefore, to systematically acquire the data as shown in line (6) of Algorithm 1, we collected the dataset under two different situations, which are (i) High Density Traffic (HDT) scenes that contains multiple objects in an image and (ii) the Low-Density Traffic (LDT) scene that contains only one class per image, with zero overlaps. For improved training, the images of the LDT and the HDT dataset are placed separately.

**The LDT Scenes:** This dataset was gathered from daily real-time traffic places, for example open parking lots, less crowded roads, and places with fewer crowds. The objective of assembling this dataset is to separately train the model on each class. We collected a total of 600 images from three classes, which are cars, motor cycles, and pedestrians. Example images of the few of the LDT images are shown in Figure 3a.

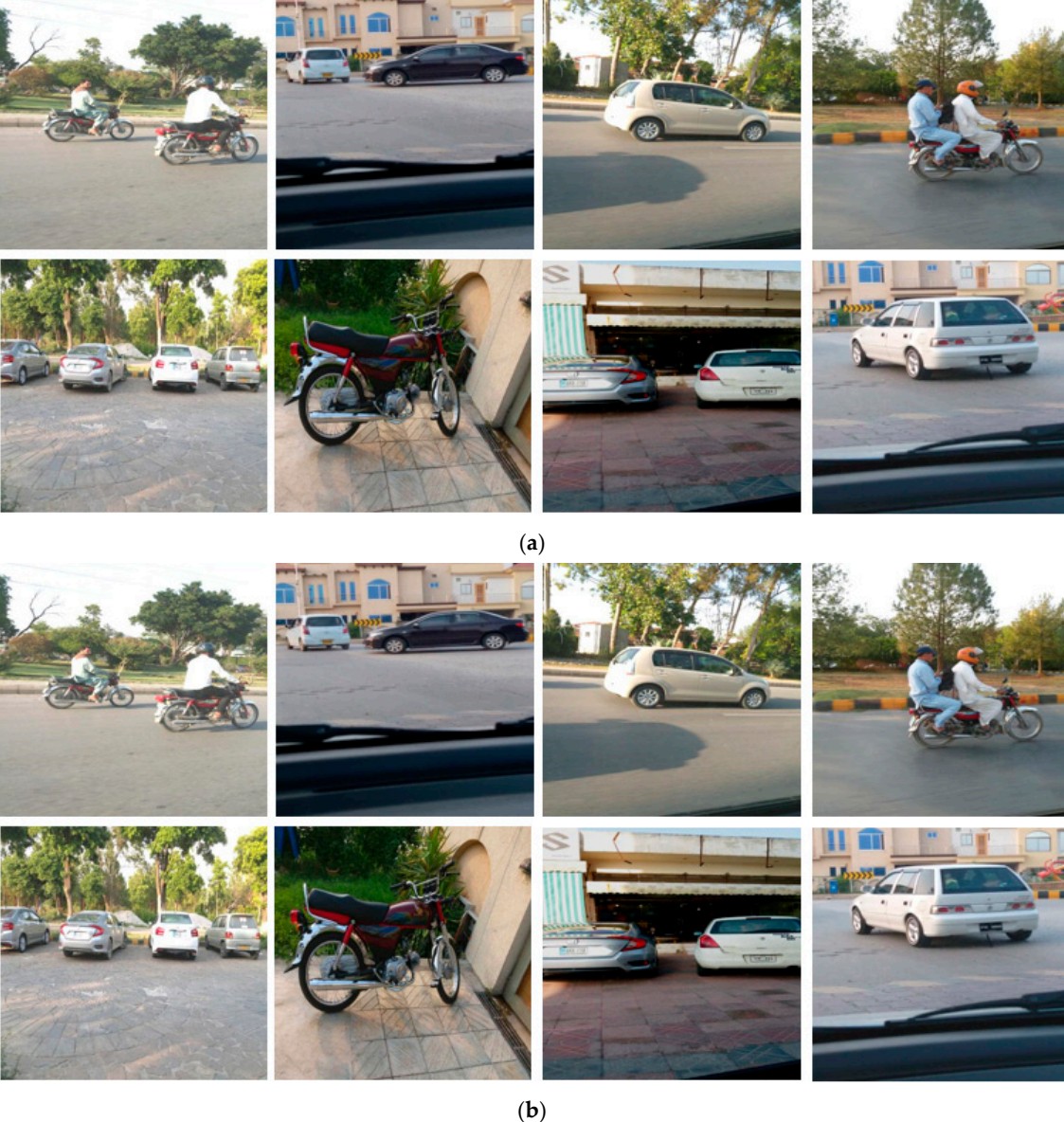

**Figure 3.** Sample images of our collected dataset: (**a**) Low density traffic scenes and (**b**) High density traffic scenes.

**The HDT Scenes:** This dataset was collected in congested places, for example public parking lots, big shopping malls, main highways, and places near main traffic sign boards. We gathered a total of 1800 images of the aforementioned classes. We also collected this dataset by thinking about crucial factors, for instance varying illumination, partial/full/long term occlusions, along with collections of objects regardless of size, scale, shape, or appearance. A few such sample images are shown in Figure 3b. The statistics of both the low- and high-density dataset along with each class annotations are described in Table 2.

**Table 2.** Summary of our collected dataset images.

| Dataset | | HDT Dataset | LDT Dataset |
|---|---|---|---|
| Source images | | 1800 | 600 |
| Annotations | | 15,618 | 903 |
| **Classes** | Car | 8457 | 655 |
| | Motorcycle | 4136 | 136 |
| | Person | 3025 | 112 |
| **Total** | | **3036** | **2406** |

**Video Dataset:** Along with the images, we also gathered a video dataset from the different locations of main highways, such as crossway bridges. A few of the sample images of our collected video dataset are shown in Figure 4a. It can be seen that our collected dataset has different types of vehicles that appear in the image. Moreover, the vehicles' resolution also varies. Collecting such a diverse dataset helps us to develop a robust, reliable, and accurate vehicle detection method, which we believe can be used in any real-time application.

### 3.2. Data Annotation

It is the proper procedure to label the classes of the datasets to achieve reliable vehicle detection in later stages. This data annotation is an important step for good training of the CNN model so as to get promising results.

As shown in line (7) of Algorithm 1, we have used Label Image Tool (https://github.com/heartexlabs/labelImg (accessed on 12 January 2023)) (LIT) to label and annotate the image dataset. To use the LIT tool, we upload the image dataset to the LIT, which reads the images. Later, we manually assign a bounding box for each object present in the image as shown in Figure 4b. It is evident that for the HDT category, there are several bounding boxes on a single image. However, for the LDT scenes, there are fewer bounding boxes. These bounding boxes specify the label of the respective class, such as vehicle or motorbike. Every object present in the image is manually labeled, which is indicated by bounding boxes. The overall dataset is then divided into three classes, which are cars, motorcycles, and pedestrians. Readers are referred to the LIT link, which is provided at the bottom of this page, which offers detail about the LIT usage.

For the video dataset, the annotation is some way bit extensive. To do it quickly, we used Dark Label Tool (https://github.com/darkpgmr/DarkLabel (accessed on 12 January 2023)) (DLT) as it consumes less time as compared to the LIT module. The DLT automatically divides the uploaded video dataset into frames, for instance frames of 10 s into 360 frames. These frames are now interpolated, in which the first frame draws a bounding box around an object, and the last frame draws the bounding box around the same object. Hence, all the objects in between the 10 s have been annotated and labelled according to the specified classes. Readers are referred to the DLT link, which is provided at the bottom of this page, which detail about the DLT usage, along with more facilities provided therein.

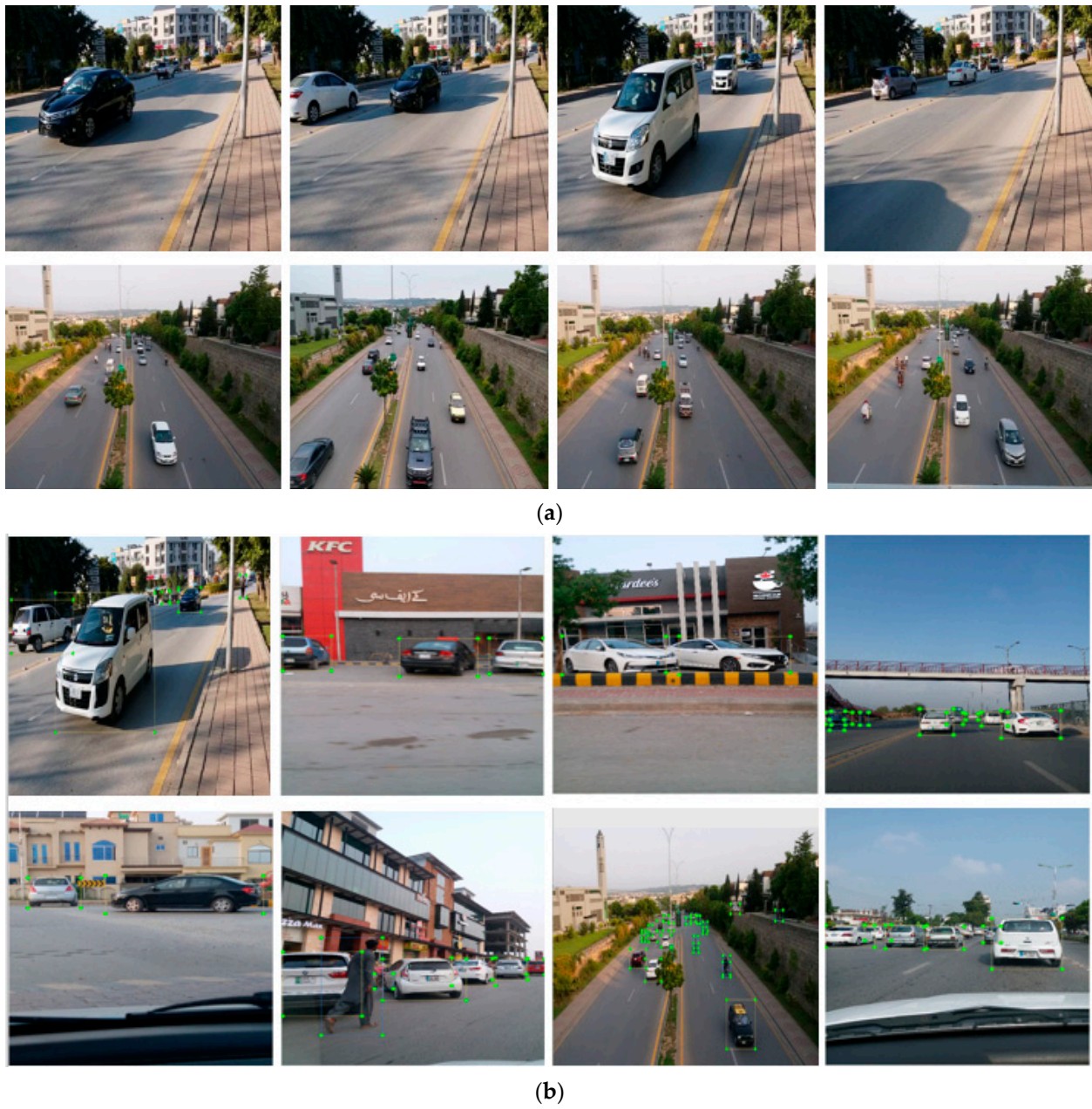

**Figure 4.** Sample images of our collected dataset: (**a**) video dataset and (**b**) annotated images.

### 3.3. Data Augmentation

To increase the data features to obtain better results, data preprocessing is the building block of deep learning-based algorithms. We know that real world datasets might be contaminated with noise. Many times, these datasets are inconsistent, or some things may be missing. Sometimes, uneven and unbalanced classes appear. As can be seen in lines (9) to (18) of Algorithm-1, data preprocessing and augmentation is analyzed. As shown in Table 3, we preprocessed our dataset in distinct steps. One is to get the same size of each image of the HDT, the LDT, and video datasets. We make the dataset of 416 × 416 pixels resolution of each image and video. Then the dataset is split into train, test, and validation set. To split the dataset, we used the RoboFlow (https://public.roboflow.com/ (accessed on 12 January 2023)) (RFW) tool as described below.

**Table 3.** Dataset statistics after augmentation.

| Dataset | Classes | Classes | Training | Validation | Testing |
|---|---|---|---|---|---|
| HDT data | 3 | car, motorcycle, | 3685 | 356 | 177 |
| LDT data | 3 | and person | 1260 | 120 | 60 |
| COCO 2017 | 80 | car, motorcycle, person, dog, table, and horse | 118,287 | 5000 | 40,760 |
| | **Total** | | **123,232** | **5476** | **40,997** |

**The RFW:** this tool hosts free public computer vision datasets in many popular formats. The RFW provides a streamlined workflow to identify edges of various objects in several iterations. With each iteration, the detection models become smarter and more accurate. We used the RFW tool to fragment the entire dataset into train, validation, and test sets. In this study, we keep the split ratio as 7:2:1 that is the image dataset of both categories and the video dataset has been divided into 70% train, 20% validation, and 10% test sets. Training a model on small number of images could result in overfitting [26]. Moreover, it also results in poor generalization despite the fact that the training results are good enough. However, the testing accuracy drops down and the model classifies the samples into one class. In short, the training accuracy is high, but the validation accuracy drops down. To overcome this issue, data augmentation is used, which modifies the data using different techniques and increase the samples of the dataset. Through empirical analysis, we applied the following augmentation techniques on our collected dataset.

*Cropping:* In this stage, we crop the image dataset of both categories between 0% minimum and 30% maximum zoom.

*Saturation:* To achieve better results, we change the color ranges of images of both categories and the video as well. In this study, we saturate images ±25%.

*Brightness variation:* The brightness of the image dataset of both categories has been carefully varied. We darkened and brightened the images ±25%. After applying the data augmentation technique to the image dataset, it is ready to use in a model for object detection. The statistics of the dataset after augmentation are shown in Table 3. As can be seen, we have a total of 123,232 training images, 5476 validation images, and 40,997 test images.

### 3.4. The YOLO-v5

The YOLO-v5 is one of the latest models to obtain reliable object detection in the YOLO family [26]. YOLO-v5 has four more types, which are, YOLO-v5s, YOLO-v5m, YOLO-v5l, and YOLO-v5×. All of these types differ in size and inference time. The size ranges between 14MB to 168MB. The YOLO-v5 surpasses other conventional object detection procedures mostly in terms of detection accuracy. Moreover, the YOLO-v5 is computationally faster in comparison to its companion YOLO family-based algorithms. As shown in Algorithm 1, the YOLO-v5 is used in this study from lines (19)–(35). There are three main architectural blocks in the YOLO-v5 as discussed below [26].

*Backbone:* In the YOLO-v5, the Cross Stage Partial (CSP) networks are used as a backbone to extract important features from the given input image. Figure 5 lists the details of the backbone modules that are embedded therein.

*Neck:* The feature pyramid is constructed with the PAN for features accumulation. The features are then passed to head. Figure 5 lists the details of the Neck module along with the necessary details, which are implanted therein.

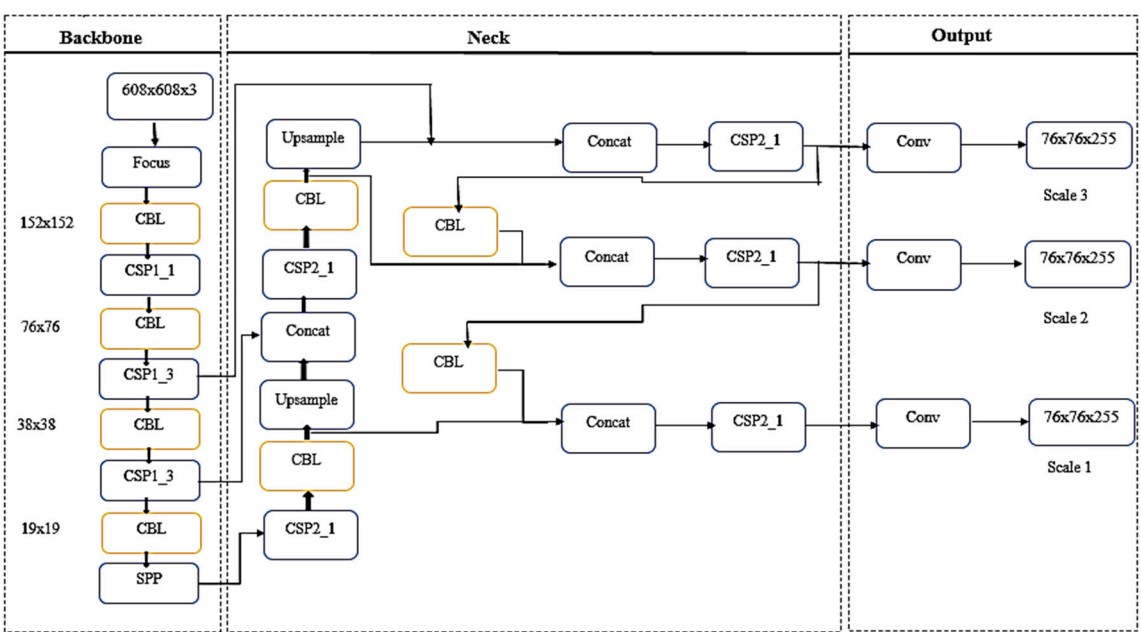

**Figure 5.** YOLOv5 schematics used in our work.

*Head:* In this block, the predictions are generated with the help of anchor boxes that ultimately achieves object detection. YOLOv-5 is made more intelligent through a transfer learning mechanism, which is shown in Figure 6, in which an input dataset is processed by the convolutional layers. That in return feeds to the FC layer. Later, our test datasets are processed by pretrained network that yields the final output through the FC layers.

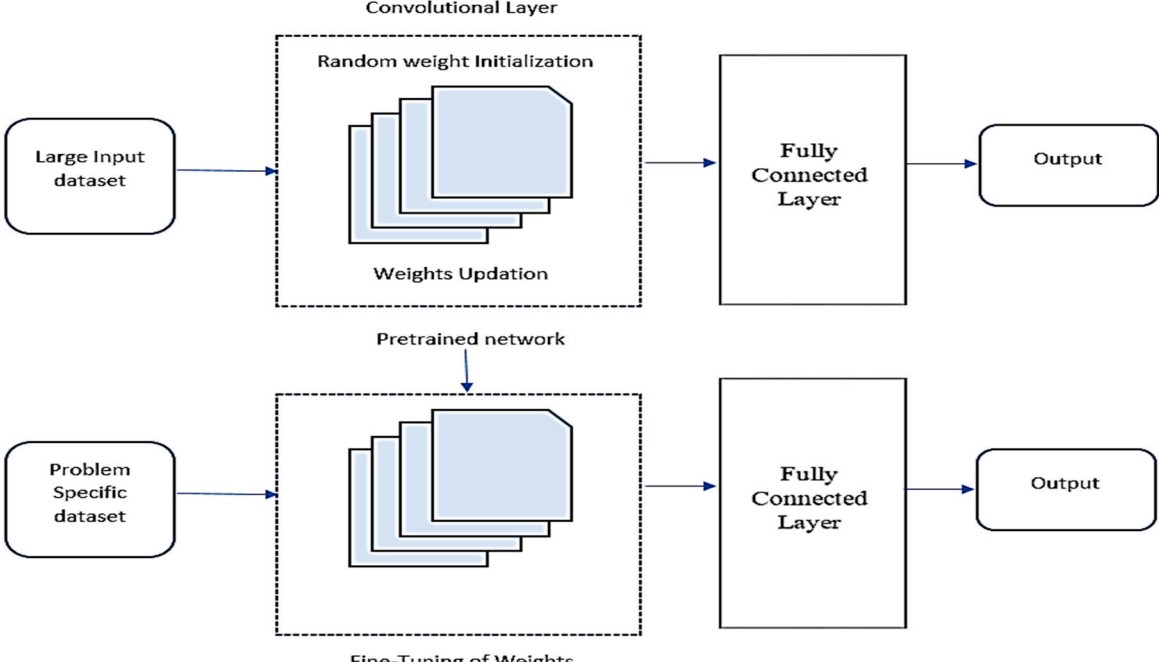

**Figure 6.** Employed transfer learning in the YOLOv5 architecture.

Therefore, to solve challenging issues in the vehicle detection domain, this above-described strategy introduces a prediction head in the YOLO v5 detection architecture. The introduced transfer learning or the prediction head is generated through random initialization of weights and the fine-tuned network. Moreover, to make the prediction head better detect various vehicles, YOLOv5 architecture also adds a CSP, CBL, and CBS

section, and up-samples the fused feature map to generate a larger feature map to detect variations in vehicles appearances.

### 3.5. Training Strategy

After the data is acquired and processed along with annotations, training and validation datasets are passed to the YOLO-v5s algorithm. For training, we select different parameters, such as batch size, epochs, and image resolution. Training path, testing, and validation dataset are given to the algorithm. We notice that if we train our model from scratch, then we have to initialize it with some random weights. Therefore, we used pre-trained COCO weights for our model training as it saves a considerable amount of time and makes computations easy. Using pre-trained YOLO-v5 model, we get the best weights after transfer learning. Moreover, we used default layers and anchors, as we are utilizing the initial weights of the COCO dataset. We also employed the COCO dataset as a benchmark to train our custom dataset. Furthermore, we have also varied the batch sizes as 5, 10, and 20. We have also changed the epochs to 100, 300, and 500. In this study, the values of confidence $\in [0.4 \sim 0.6]$. After the training phase is done, we use the best weights to detect objects on the dataset. Lastly, we obtain the values of predicted labels and the test images with the bounding boxes with confidence values. The collected dataset and the trained model along with the manuscript will be made publicly available (http://research.cuiatd.edu.pk/secure/ResearchGroups/comsatsresearchgroups.aspx (accessed on 12 January 2023)). In the next section, we present and discuss the simulation results in detail.

### 3.6. Evaluation Criteria

The following criteria are used to measure the robustness of our developed vehicle detection method:

$$Precision = \frac{True\ Positive\ Cases}{Total\ Positive\ Predictions} \qquad (1)$$

$$Recall = \frac{True\ Positive\ Cases}{Total\ Cases} \qquad (2)$$

Similarly, mean Average Precision (*mAP*), the average value of Precision, is also computed for the value of Recall over 0 to 1. The mAP is usually applied in object detection algorithms and is shown mathematically below.

$$mAP = \int_0^1 P(R)dR \qquad (3)$$

## 4. Simulation Results

This section presents the detailed simulation results. Extensive experiments are carried out on Google Colaboratory (Colab) platform. The Google Colab provides Intel Xeon CPU with a clock speed of 2.3 GHz and up to 16 GB of RAM. Moreover, the Google Colab also provides NVIDIA K80 or T4 GPU. We use Python V3.6 as a simulation tool for different vehicle datasets as described in subsequent sections. To investigate the performance of vehicle detection methods on different datasets, we select 14 state-of-the-art vehicle detector evaluations and comparisons with the proposed method in terms of accuracy and execution time. All of the compared approaches have been trained on the same training data from each of the PKU, COCO, and DAWN datasets.

### 4.1. Analysis on the PKU Dataset

The PKU dataset is a collection of diverse vehicle images that are captured under diverse conditions [27]. As shown in Table 4, that this dataset contains a total of 3977 diverse vehicle images. The developers of the PKU dataset divided the vehicles into five distinct and different categories, which they refer as G1, G2, G3, G4, and G5. Out of 3977 vehicle images, the PKU dataset also contains a total of 4263 visible license plates

whose pixel resolution varies from 20~62 pixels, which are captured therein. Figure 7 shows a few of the vehicle detection results of our proposed YOLO-based method on all of the five categories of the PKU dataset. As can be seen in the first three rows of Figure 7, for the G1~G3 categories, the proposed method locates all the vehicles in the input images. For the G4 category as shown by the fourth row in the Figure 7, it is obvious that the proposed method is able to locate vehicles that just expose their front bonnet. Moreover, the PKU–G4 category also contains extreme reflective glare. It is always very challenging for any detection algorithm to perform accurately under such circumstances. However, as can be seen, the proposed method handles the aforesaid scenario effectively.

**Table 4.** The PKU dataset description.

| Category | Vehicle Conditions | Input Image Resolution (pixels) | No. of Images | No. of Plates | Plate Height (pixels) |
|---|---|---|---|---|---|
| G1 | Cars on roads; ordinary environment at different daytimes; contains only one vehicle/license plate per image. | 1082 × 728 | 810 | 810 | 35–57 |
| G2 | Cars/trucks on main roads at different daytimes with sunshine; only one vehicle in each image. | 1082 × 728 | 700 | 700 | 30–62 |
| G3 | Cars/trucks on highways during night; one license plate per image. | 1082 × 728 | 743 | 743 | 29–53 |
| G4 | Cars/trucks on main roads; daytimes with reflective glare; one license plate in input images. | 1600 × 1236 | 572 | 572 | 30–58 |
| G5 | Cars/trucks at roads junctions with crosswalks; several vehicles per image. | 1600 × 1200 | 1152 | 1438 | 20–60 |
| **PKU dataset** | | | **3977** | **4263** | **20~62** |

Moreover, the proposed method also performs well on the G5 category, which contains multiple vehicles per image. It can be seen that for different view angles along with the partially occluded vehicles, the proposed method performs well and in most of the instances detects all such vehicles. Figure 7 also reveals that the PKU-G3, G4, and G5 pose a challenge to any detection algorithm due to the fact that the illuminations change abruptly. A few of the images shown in the 3rd, 4th, and 5th rows in Figure 7 have a background that is dark black, or in which the head lights of the vehicles are turned on. In such cases, the proposed method performs well and up to task by locating all the vehicles that appear therein.

Table 5 lists the comparison of the proposed method on the PKU dataset with fourteen other methods. Since we collected most of the data from Pakistani cities, for a fair comparison we tested the methods reported in [28–31] and [33–42] on the whole PKU dataset along with our developed method. Table 5 lists the detailed results with important observations.

- From Table 5, it is evident that all the compared methods except [33,35–37,41] and [42] yield 100% detection accuracy on the PKU–G1~G3 categories.
- On the G4 category, the proposed method ranks 3rd among all the compared fourteen methods in terms of detection accuracy. On the other hand, on the G5 category, our developed method outperforms all the compared methods. The PKU G5 is a challenging category due to fact it contains multiple vehicles per image and also contains several disguising crosswalks that pose a threat to any vehicle detection algorithm.
- From Table 5, we also observe that for G1, G2, and G3 categories, the methods developed in [28–30,38,40] produce 100% vehicle detection result. Our proposed method yields 99.94% vehicle detection accuracy on the G1 category and 100% for the G2 and G3 categories. Therefore, we observe that methods shown in Table 5 have solved the challenge of vehicle detection on these three categories as most of them yield at least 99% accurate vehicle detection.

- Overall, on the PKU dataset, the proposed method ranks 1st at achieving vehicle detection in terms of the mAP as listed in Table 5. The method by [40] ranks 2nd by yielding 99.86% accurate vehicle detection accuracy. The works developed in [30,31] also yield slightly over 99.75% vehicle detection accuracy. In addition, the methods shown in Table 5 report over 97% detection accuracy, which we believe is encouraging in solving real-world traffic problems.
- To best of our knowledge, we observe that vehicle detection challenge is almost solved on the PKU dataset. However, we observe that non-uniform illuminations or high glare at the night could still affect vehicle detection accuracy. Similarly, the researchers who aim to solve the other object detection problems, such as license plate detection or recognition, may need to perform additional preprocessing or postprocessing to achieve reliable detection results.

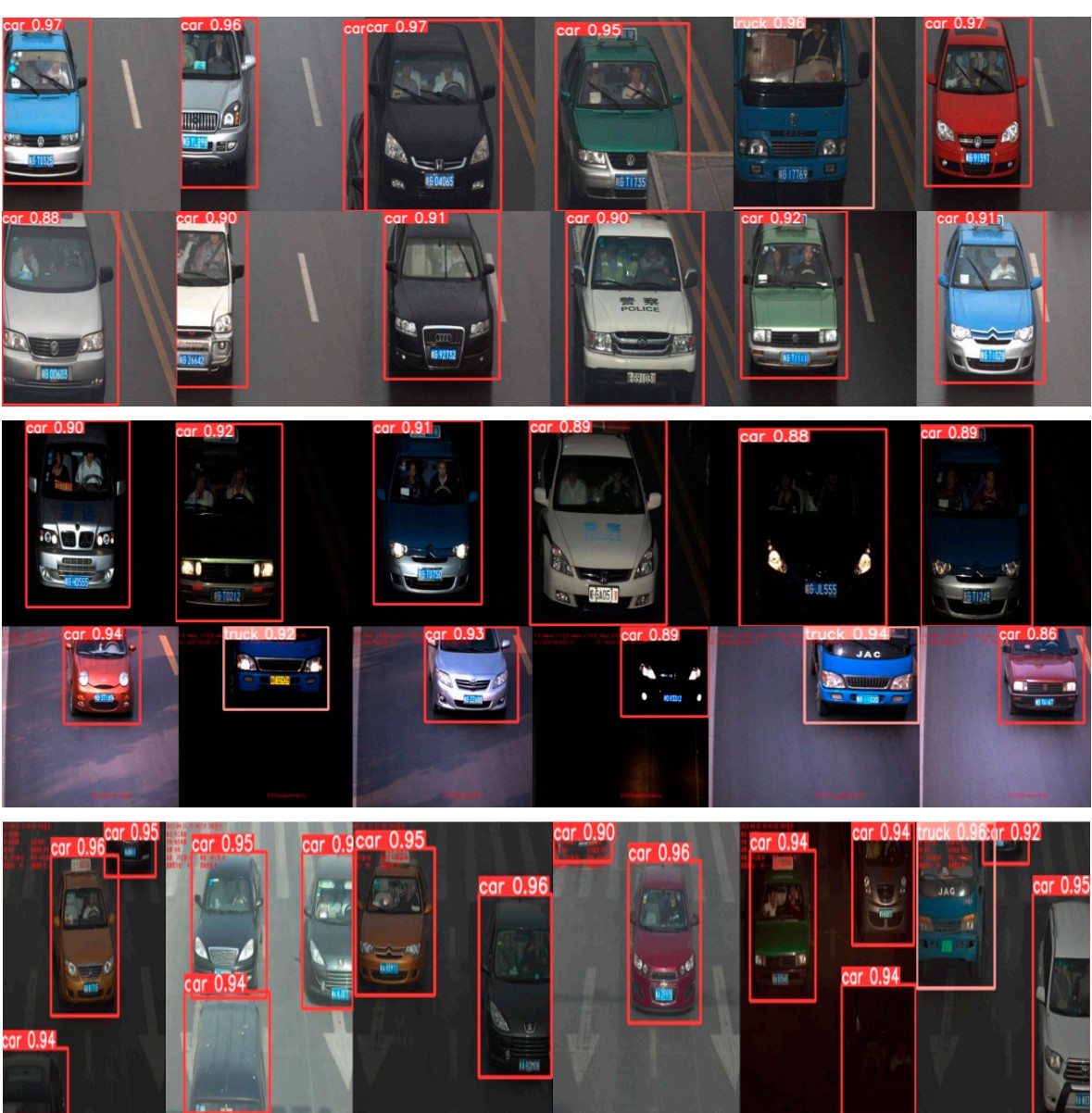

**Figure 7.** Vehicle detection results on the PKU dataset from G1 to G5 categories

**Table 5.** Vehicle detection comparison (%) on PKU dataset.

| Ref. | G1 | G2 | G3 | G4 | G5 | mAP (%) |
|------|------|------|------|------|------|---------|
| [28] | 100 | 100 | 100 | 98.96 | 99.13 | 99.61 |
| [29] | 100 | 100 | 100 | 99.73 | 99.21 | 99.78 |
| [30] | 100 | 100 | 100 | 99.70 | 99.10 | 99.76 |
| [31] | 100 | 100 | 99.40 | 99.74 | 98.96 | 97.74 |
| [33] | 99.00 | 99.00 | 98.70 | 98.00 | 98.90 | 98.72 |
| [34] | 100 | 100 | 100 | 99.00 | 96.50 | 99.10 |
| [35] | 100 | 100 | 99.00 | 99.64 | 99.06 | 98.34 |
| [36] | 100 | 100 | 99.40 | 99.74 | 98.96 | 97.74 |
| [37] | 100 | 98.50 | 100 | 99.50 | 98.10 | 99.22 |
| [38] | 100 | 100 | 100 | 99.00 | 98.00 | 99.40 |
| [39] | 99.00 | 100 | 100 | 99.00 | 98.50 | 99.30 |
| [40] | 100 | 100 | 100 | 99.80 | 99.50 | 99.86 |
| [41] | 99 | 99 | 98.50 | 98.00 | 99.00 | 98.70 |
| [42] | 98.90 | 98.50 | 98.00 | 97.50 | 96.10 | 97.80 |
| **Proposed** | **99.94** | **100** | **100** | **99.73** | **99.96** | **99.92** |

*4.2. Analysis on the COCO Dataset*

The COCO dataset is designed to detect and segment various objects that occur in their natural context [32]. As shown in Table 3, the COCO dataset contains various object images, which have been gathered from complex everyday scenes and contains common objects in their natural context. Moreover, objects in this dataset are labeled using per-instance segmentations to aid in precise object localization. Overall, the COCO dataset contains images of 91 object types with a total of two and a half million labeled instances in 328,000 images. Recently, the COCO dataset received extensive attention from researchers investigating various categories of detection including diverse vehicle shapes. Figure 8 shows the vehicle detection results of our proposed method on the COCO dataset. Clearly, Figure 5 depicts the performance of the proposed vehicle detection algorithm on various challenging images of the COCO dataset.

In most of the instances and under huge illumination variations, almost all of the different vehicles are accurately detected by the proposed methodology. We used other objects, such as motorcycles and persons during this phase. Therefore, those are also accurately located in various images in Figure 8. A few such instances can be seen in the 1st image of the 2nd and 3rd rows, respectively. Similarly, the 6th image in the bottom row of Figure 8 also depicts the object detection phenomenon. To further validate the superiority of the proposed method, a comparison with fourteen other methods is listed in Table 6 with some important observations.

- As can be seen for various image resolutions that range from $512 \times 512$ to $800 \times 800$ pixels, the proposed method ranks 1st among all the compared methods and reports the highest mAP value of 52.31%. The work reported in [42] ranks 2nd and yields a 50.40% mAp value followed by [41] with a 49.80% mAP value. Our analysis reveals that the work developed in [35,36] are also an encouraging solution for detecting various objects in the challenging COCO dataset.
- On the COCO dataset, the work reported in [31] yields the lowest (27.89%) mAp value followed by [33], whose method yields a mAP value of 29.10%. Moreover, in the current study, work discussed in [40], which uses the ResNet as a backbone, yields a mAP value of 31.80%, which in the context of current study falls on the lower side.

- The crux of this dataset is that the proposed method effectively and reliably detects miscellaneous objects that include vehicles of varying shapes, including motorbikes and jeeps. Furthermore, the proposed method also effectively handles big buses. A few such samples are also in the 1st and 3rd columns of Figure 8.

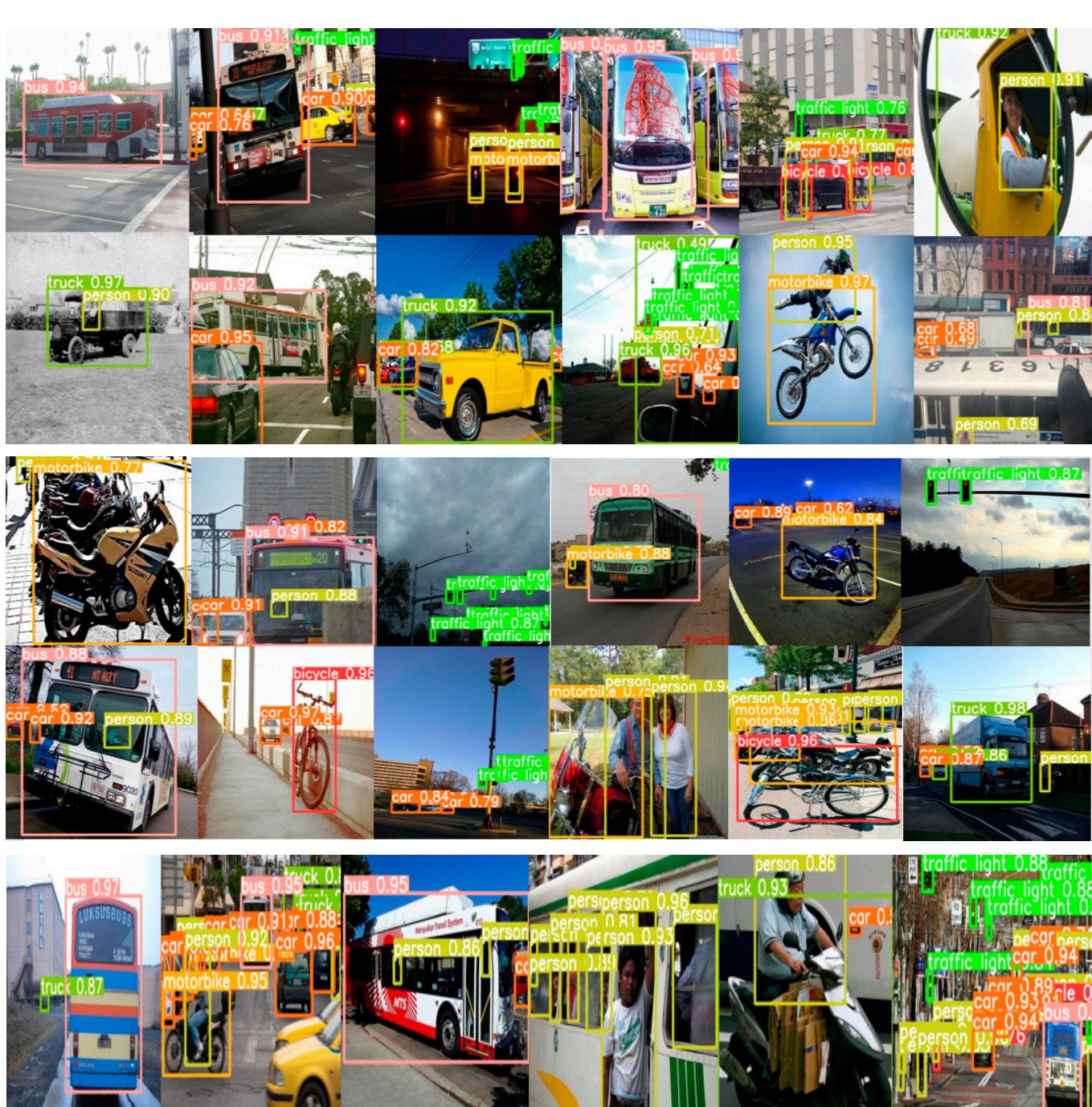

**Figure 8.** Vehicle detection results on the COCO dataset.

**Table 6.** Vehicle detection accuracy comparison on the COCO dataset.

| Ref. | Backbone | Data | Input Size | Multi Scale | mAP(%) |
|---|---|---|---|---|---|
| [28] | CSPDarkNet53 | trainval35K | $512 \times 512$ | False | 47.62 |
| [29] | CNN | trainval35K | $512 \times 512$ | False | 48.00 |
| [30] | R-CNN | trainval35K | $512 \times 512$ | False | 46.20 |
| [31] | BottlenectCSP | trainval35K | $512 \times 512$ | False | 27.89 |
| [33] | VGGNet-16 | trainval35K | $512 \times 512$ | False | 29.10 |
| [34] | ResNet-101-FPN | trainval35K | $512 \times 512$ | False | 38.30 |
| [35] | VGGNet-16 | trainval35K | $800 \times 800$ | False | 41.00 |
| [36] | ResNet-101 | trainval35K | $800 \times 800$ | False | 48.40 |
| [37] | ResNet-101 | trainval35K | $512 \times 512$ | False | 39.30 |
| [38] | CNN + SVM | trainval35K | $512 \times 512$ | False | 49.05 |
| [39] | BN + ReLU | trainval35K | $512 \times 512$ | False | 32.98 |
| [40] | ResNet-C4-FPN | trainval35K | $512 \times 512$ | False | 31.80 |
| [41] | ResNet-50 | trainval35K | $512 \times 512$ | False | 49.80 |
| [42] | SiNet | trainval35K | $512 \times 512$ | False | 50.40 |
| **Proposed** | **CSP** | **trainval35K** | $\mathbf{512 \times 512}$ | **False** | **52.31** |

### 4.3. Analysis on the DAWN Dataset

The DAWN dataset is designed to investigate the performance of recent vehicle detection methods on a broad range of natural images including adverse weather conditions. The DAWN dataset contains 1000 image of significant variation in terms of vehicle size and category along with pose variation, non-uniform illumination, position, and occlusion from real traffic environments. Additionally, it exhibits a systematic variation for traffic scenes, for instance, bad winter weather, heavy snow, sleet rain, sand, and dust storms. Figure 9 shows detailed results on fog, sand, rain, and snow situations with important observations.

- For the snow category as seen in top row of Figure 9, it is obvious that many times the vehicles are partially visible due to adverse weather conditions, such as fog that is normally experienced in severe winters in areas of various parts of the world. However, our developed method handles all such situations except the 2nd last image of front row in Figure 9, where it is obvious that the vehicle is not visible to the human eye as well.

- For a considerably rainy day as seen in second row of Figure 9, the proposed method accurately locates multiple vehicles that appear therein. In this case, the image scene variations, such as shown in the 2nd and 4th images of the second row in Figure 9 indicates that the proposed method is unaffected by such changes in the image scene. Similarly, the skyscrapers in the vehicle's background as shown in the 5th image of the 2nd row in Figure 9 also do not affect the detection ability of our developed method.

- For a sand situation as indicated in the third row of Figure 9, the proposed method detects all vehicles that appear there in such challenging conditions. In such situations, visibility is normally very low, which poses threats to most of the machine learning algorithms. Particularly, the first two images in the 3rd row of Figure 9 have intra-class scene variations, i.e., both are images effected by sand storms and yet appear differently to the human eye. Even in such cases, our developed method performs well and detects most of the instances that appear in such condition. The 3rd image in this row is quite challenging for human observers as well. However, as indicated there, our developed method handles such situations by successfully locating the vehicles that appear in such scene images.

- The bottom row in Figure 9 is a case when the scene is dominated by snow. In this case, surprisingly, the image appears neat and clean and thus results in a visually pleasing image due to the massive amount of snow which is present in the image. In this case, our developed method accurately detects and labels all the vehicles that appear therein. Particularly, the 3rd image in this row also reveals a red light along with the snow. Yet in this case, the proposed method performs well and successfully locates all the vehicles. Moreover, the last image in this row shows a few vehicles that overlap and result in partial occlusion. However, our developed method performs well in this case as well.

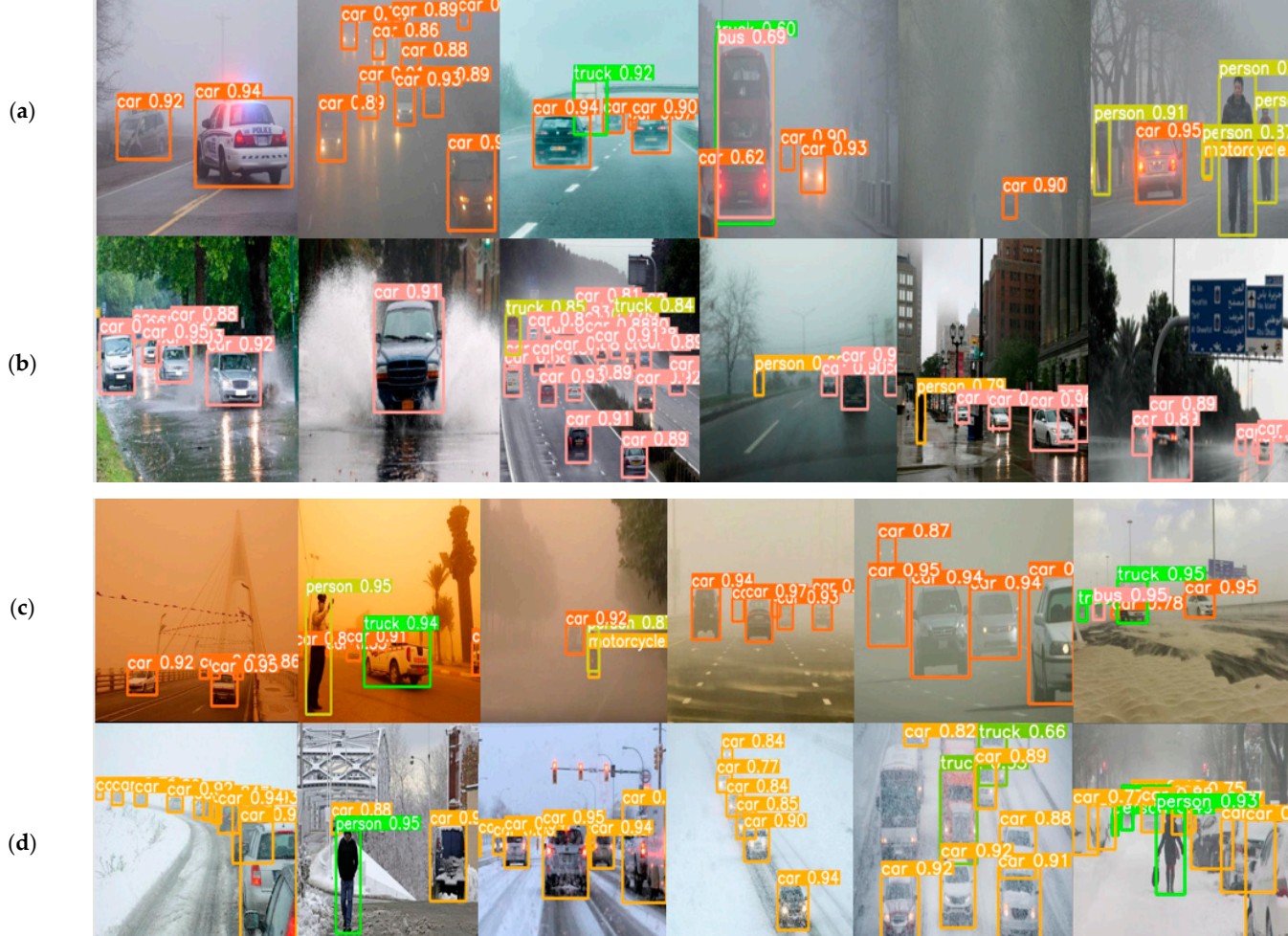

**Figure 9.** Vehicle detection results on the DAWN dataset row-wise: (**a**) fog; (**b**) rain; (**c**) sand; and (**d**) snow.

In Table 7, we compare our method with the works already described in Tables 5 and 6, respectively. A few of the observations from Table 7 are listed below.

- As can be seen in Table 7, for the fog scenario the work developed in [40] ranks 1st among 14 compared methods by yielding a 29.68% mAP value. Our developed method ranks 2nd out of all compared methods in fog situation and yields a 29.66% mAP value. In the fog situation, the work developed in [38] yields the lowest mAP value (16.50%) followed by [29] whose method yields a mAP value of 24%.

- For the rain scenario on the DAWN dataset, our proposed method and the work developed in [31] yield the highest mAP value of 41.21%. In this category, the work in [34] ranks 2nd and yields an encouraging result of a 41.10% mAP value. For the

aforesaid category, results yielded by [36,37] are also encouraging. For images that are affected by rain, the work in [38] delivers the lowest mAP value of 14.08%.

- For the snow conditions on the DAWN dataset, the work developed in [37] ranks 1st among all compared methods and slightly outperforms the proposed method by yielding a mAP value of 43.02%. For this category, our developed method yields a mAP value of 43.01%. It is important to state here that for this situation, the works in [31,33,36,37] yield almost similar results.
- For the sand condition, our method ranks 1st and outperforms all compared methods by yielding a 24.13% mAP value. On this situation, the works in [28,34,35] yield similar mAP values. For the sand situation, the work in [38] yields the lowest mAP value (10.69%).

**Table 7.** The mAP (%) comparison on DAWN dataset.

| Ref. | Backbone | Image/s | Fog | Rain | Snow | Sand |
|------|----------|---------|-----|------|------|------|
| [28] | CSPDarkNet53 | 0.085 | 26.40 | 31.55 | 39.95 | 24.10 |
| [29] | CNN | 0.085 | 24.00 | 21.10 | 38.32 | 23.80 |
| [30] | R-CNN | 0.085 | 27.20 | 21.30 | 28.30 | 18.00 |
| [31] | BottlenectCSP | 0.085 | 29.31 | 41.21 | 43.00 | 24.02 |
| [33] | VGGNet-16 | 0.085 | 23.40 | 24.60 | 37.90 | 15.83 |
| [34] | ResNet-101-FPN | 0.085 | 28.95 | 41.10 | 43.00 | 24.09 |
| [35] | VGGNet-16 | 0.085 | 23.10 | 27.65 | 34.00 | 24.10 |
| [36] | ResNet-101 | 0.085 | 29.70 | 40.10 | 43.00 | 23.99 |
| [37] | ResNet-101 | 0.085 | 28.10 | 40.40 | 43.02 | 24.10 |
| [38] | ResNet-101-FPN | 0.085 | 16.50 | 14.08 | 15.38 | 10.69 |
| [39] | Hourglass-104 | 0.085 | 25.08 | 19.14 | 23.18 | 17.38 |
| [40] | ResNeXt-101 | 0.085 | 29.68 | 30.32 | 33.93 | 24.00 |
| [41] | ResNet-101-FPN | 0.085 | 28.83 | 27.68 | 30.19 | 24.03 |
| [42] | VGGNet-16 | 0.085 | 26.45 | 20.09 | 27.92 | 11.31 |
| **Proposed** | **CSP** | **0.0085** | **29.66** | **41.21** | **43.01** | **24.13** |

## *4.4. Computational Complexity*

We evaluate the computational complexity in terms of the time consumed to yield the vehicle detected output image. While evaluating the computational complexity of the methods listed in Figure 10, we manually vary the test image size from 512 × 512 pixels up to 1600 × 1236 pixels on all the three datasets compared in this study. In addition, all the times shown in Figure 10 are the mean execution time on all the three datasets to process a single image and yield the output image.

Moreover, in Figure 10, we compare the execution time of 14 state-of-the-art vehicle detection methods with the proposed method. It can be seen that the work of Liu [34] is computationally more expensive than all of the compared methods. Clearly, the proposed method is computationally most economical and consumes slightly more than 0.50 s to yield a vehicle detected output image. Furthermore, the works reported by Wu et al. [31], Law et al. [39], and He et al. [40] consume nearly 1 s to yield the output image with detected vehicles.

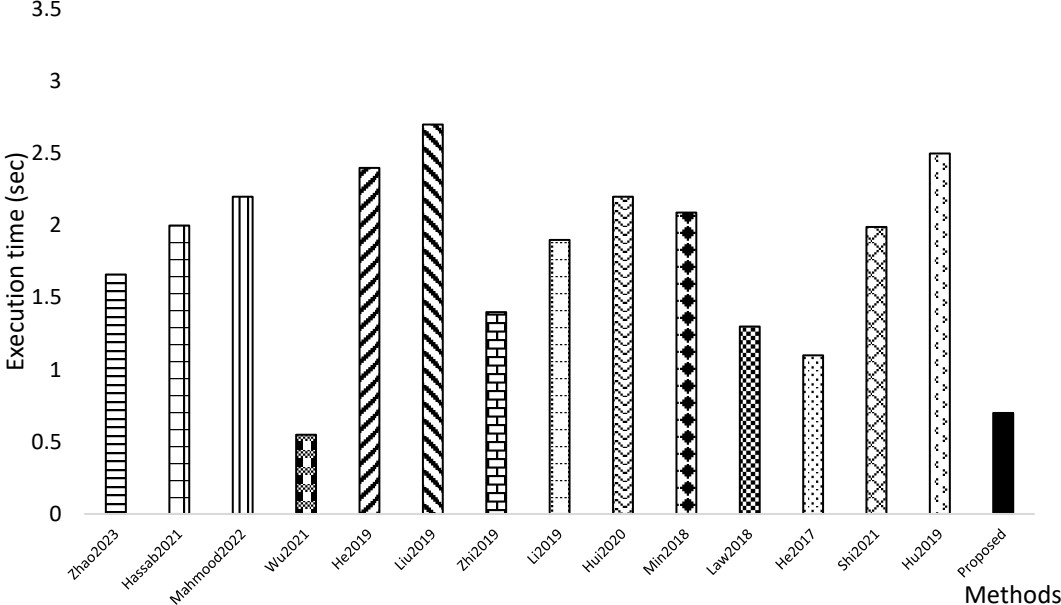

**Figure 10.** Computational complexity comparison with fourteen other approaches [28–31,33–42].

*4.5. Discussion*

Although the analysis presented above familiarizes the readers with the feasibility of our developed method to detect diverse vehicles under diverse range of environments, the discussion below will give further insight to the readers.

- Methods compared in this study are state-of-the-art object detectors. We observed that specific method performs well on a specific dataset but are challenged by other datasets. For instance, the work developed in [28] investigates BIT-Vehicle and UA-DETRAC datasets only. These datasets mostly contain high quality frontal view of the different vehicles with image resolution of 1920 × 1080 to 1600 × 1200 pixels. In contrast, the method proposed in current study explores three different datasets that have variations, such as different road conditions, varying weathers, or complex backgrounds. Moreover, the study presented in this manuscript also explores the detection ability of this method on three standard and publicly available datasets.
- The works discussed in [29,34] mostly focus on KITTI and the DAWN datasets that contain the variations as described earlier. However, we also explore their detection ability on five different classes of the PKU dataset that contain huge road and traffic variations along with our proposed method. This will essentially provide a nice baseline to beginners and researchers to develop their specified tasks.
- The work reported in [30] investigates the generic PKU dataset in its five distinct categories. However, this study further explores the detection capability of [30] on the COCO and the DAWN datasets. Moreover, the detection accuracy of the method proposed in this study provides a fair insight into vehicle detection in various scenarios.
- The method developed in [31] examined the CARLA dataset, which we believe is a limited and relatively small vehicle dataset. The findings presented in this study extend the detection capability of this method to three other datasets. In addition, its detection comparison with the proposed method and several other techniques provides much detailed insight about issues in the vehicle detection domain.
- In [33,37], the PASCAL VOC 2007 dataset is explored only. Moreover, work in [33] also analyzes the subdomain of the COCO dataset to show the detection of trains only. In contrast, this study explores the detection capability of [33] on various vehicle classes of the COCO dataset along with the PKU and DAWN datasets. Moreover, the detailed comparison provided in the earlier sections provides a fair baseline to the research community. Furthermore, the work in [37] explored the PASCAL dataset

that already contains annotated images of various objects. This study further expands the detection capability of this method to three different vehicle datasets. Finally, the detailed analysis and comparison provided in earlier section hints towards additional modifications of this algorithm.

- The works in [35,36,39,40] were validated on the MS COCO dataset to detect various objects. The experiments reported in this study extend the detection analysis of the aforementioned approaches to PKU and DAWN datasets as well. Since our method also explores the vehicle detection on these datasets, it will be convenient for researchers and practitioners to choose the appropriate algorithm for their specified applications. Moreover, the work listed in [40] reports the detection of various objects, such as pedestrians, statues, or animals. However, this study reports the detection ability of this algorithm on actual and real-world vehicle images along with several other approaches.

- In [38], the PETS2009 and the changedetection.net 2012 datasets are explored. Results analyzed in their study are mostly standard high quality frontal view images of mono-color cars running on a main highway. In contrast, the analysis presented in this study explores the detection ability this method on different datasets on multiple styles of vehicle and on differently color cars. Moreover, this study also investigates the detection ability of this method on varying illuminations and weathers along with different road conditions.

- The study in [41] analyzed the DLR Munich vehicle and VEDAI datasets. In their study, mostly high-quality aerial vehicle images are analyzed. Few of these are running on roads, while several parked vehicles are shown. However, our study also reports the use of this method on actual daily life vehicle images from three publicly available datasets. We are optimistic that detailed analysis and comparisons provided in this manuscript will be handy for the research community to modify any algorithm for their specified tasks.

- Finally, in [42], the KITTI and the LSVH datasets were explored. Results reported in this study are mostly vans, cars, and bus that run on the main highways. However, our study reports the detection ability of this method on varying illuminations, different weathers, and challenging road conditions from three publicly available datasets. We believe that the analysis provided by our developed method and the detailed comparison listed in this manuscript will provide further insight to the research community.

- All of these are useful efforts to solve and automate the vehicle detection problem under varying conditions. For each of the datasets mentioned above, these methods perform well. One of the objectives of the current study was to test and analyze all of the fourteen methods compared in this paper on standard PKU, COCO, and DAWN datasets. The main reason to choose PKU, COCO, and DAWN datasets is that these datasets contain real world and challenging images. For instance, the PKU dataset has five distinct categories that range from normal images to dark night images along with night glare. Similarly, this dataset also contains multiple images that appear in the input along with partial occlusions and different road conditions. Similarly, as mentioned in Section 4, the COCO dataset is also a huge dataset and contains a diverse range of objects. Moreover, the DAWN dataset also contains various real-world situation, such as fog, rain, snow, and the sand. An evaluation of fourteen different methods on these three datasets will be a fair guideline for researchers and beginners to develop, implement, or modify any algorithm for their specified applications.

- Out of the datasets that are investigated in this study, we find the DAWN dataset a bit more challenging than the others. The main reason is the inclusion of images in challenging conditions, such as fog, rain, contaminated with sand, or snow. Our study indicates that the sandy images reduce the scene visibility and ultimately reduce the detection accuracy of a detector. The 1st image in the top row in Figure 11 depicts such conditions in which very low vehicle detection is achieved. Similarly, as shown in the 2nd image of the top row of Figure 11, low vehicle detection is observed during

a rainy night when the head lights of the vehicle are also turned on. In this case an electricity pole also appears, which results in partial occlusion that ultimately results in reduced object detection.

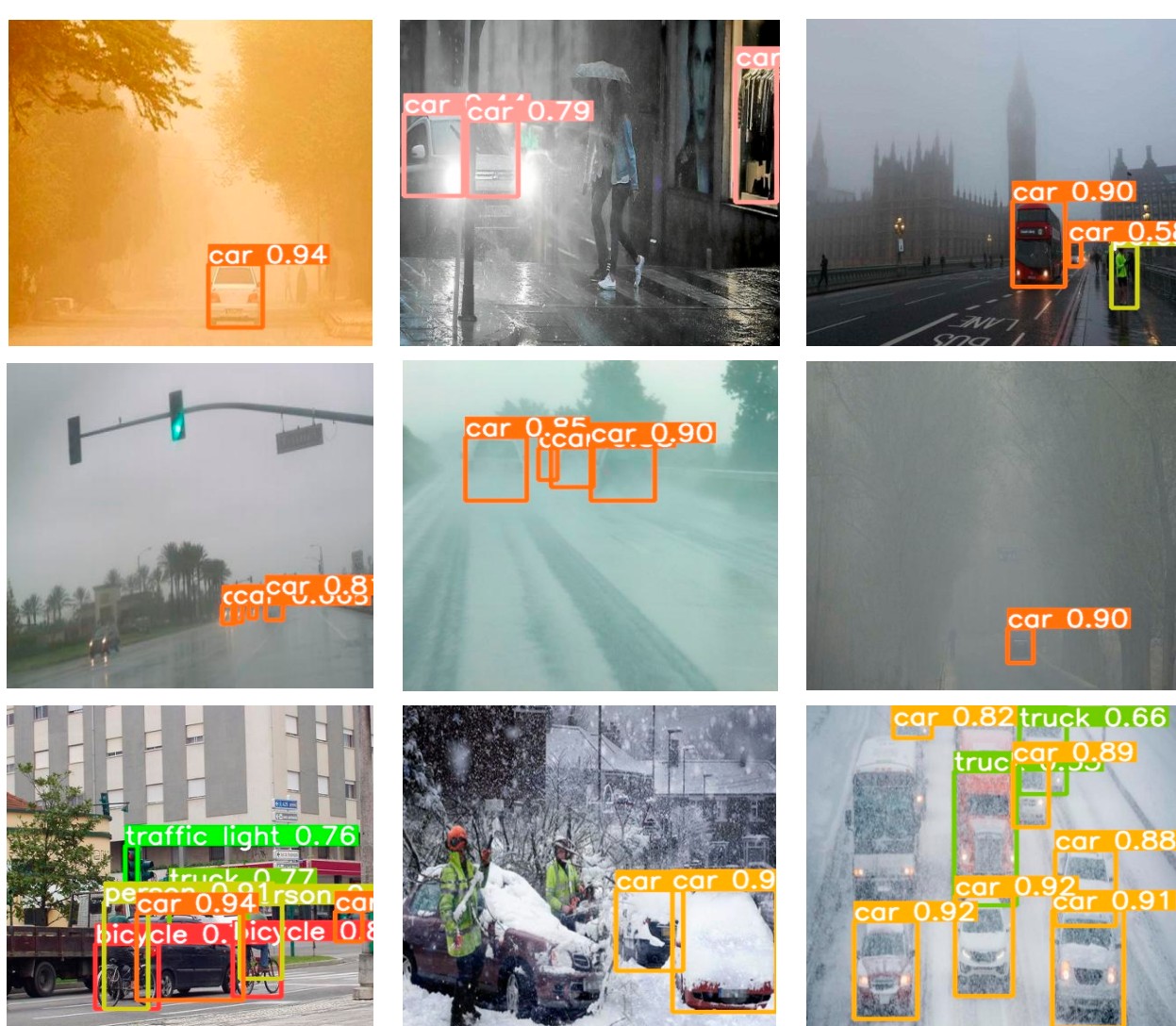

**Figure 11.** Sample images of low vehicle detection from compared datasets

- We observe that our proposed method still needs to perform well in different situations, such as when the scene is contaminated with the snow storm or blizzard as shown in the 2nd row of Figure 11. In such cases, background noise dominates results in low visibility. In this scenario, a Retinex-based image enhancement scheme might be useful. For such a scenario, we suggest that an image dehazing-based enhancement could also be effective. We are optimistic that this proposed solution will essentially enhance the object and image scene, which will later make life easier for any of the vehicle detectors deployed. Ultimately, the application of image enhancement technique will significantly increase the detection ability of object detector.
- For images where snow is dominant, image appears overly white, which also decreases the detection accuracy of state-of-the-art object detection methods. In this case, image contrast correction might produce the desirable results. In many cases, the occlusions on the road also pose a threat to the detector, which ultimately results in false detections. In such cases, an occlusion handling method could also be used to reliably detect any object.

- For all of the aforementioned discussion, Figure 11 shows a few of the sample images where our developed method struggles. In images shown in Figure 11, our method either yields a very low vehicle detection rate or produces false detections. Therefore, future research could also focus on few of the cases as shown in Figure 11.

### 4.6. Final Remarks

Detailed analysis discussed in this paper indicates that vehicle detection has been an active research field in recent years. From providing early warning signals and monitoring up to exercising control, there are several examples of major research in intelligent vehicle detection. This paper presented a detailed analysis on vehicle detection on three publicly available dataset. For the task of vehicle detection, YOLO-v5-based architecture was used. To make the YOLO-v5-based architecture more intelligent and flexible, a transfer learning methodology was introduced. Detailed analysis indicated that the proposed approach performed well on challenging datasets.

In addition, a detailed comparison of the proposed method was carried with fourteen recent state-of-the-art approaches. We are optimistic that this study will be a fair guideline for beginners and practitioners to modify or use any detector for their desired tasks or applications. Below we list the final summary of developed vehicle detection method on three datasets.

**PKU:** On this dataset, in the G1 category, the proposed method yielded mAP of 99.94%. In the G2 and G3 categories, the proposed method yielded 100% vehicle detection mAP. In the G4 and G5 categories, the proposed method yielded 99.73% and 99.96% mAP, respectively. Overall, on the PKU dataset our method yielded 99.92% vehicle detection accuracy. Out of the fourteen compared methods on the PKU dataset, the proposed method ranked 1st among all compared approaches.

**COCO:** On this dataset, with image resolution of $512 \times 512$ pixels, the proposed method yielded 52.31% mAP values and ranked 1st among all the compared methods.

**DAWN:** This dataset contains four prominent sub classes, which are fog, rain, snow, and sand. On images that were affected by fog, our proposed method yielded a mAP value of 29.66% and ranked 3rd out of fourteen compared methods in this category. Meanwhile, for images that contained rain, our developed method produced a mAP value of 41.21% and ranked 1st along with [31] among all compared works. For images that contained snow, our method yielded a 43.01% mAP value and ranked 2nd among all compared works. In this class, the work developed in [37] ranked 1st by yielding a 43.02% mAP value. For images that contained sand, our developed method yielded a 24.13% mAP value and ranked 1st among all methods. In this class, the work developed in [28] also produced a par result by yielding a mAP value of 24.10%.

## 5. Conclusions

This paper discussed an accurate, fast, and robust vehicle detection method based on the YOLO-v5 architecture. To develop a robust object detection algorithm, transfer learning was performed. The proposed object detection method was tested on three publicly available datasets, which are the PKU, COCO, and DAWN datasets. Simulation results demonstrated that the proposed method is effective at handling various challenging situations, such as night, rainy, and snow conditions. The proposed method significantly elevated the accuracy and operational efficiency. In addition, the detection technique proposed in this research can additionally be relevant to a large number of real time applications. However, the only caveat is that a giant quantity of data is required for training of the detection model. The YOLO-vs-based vehicle detection method discussed in this paper achieved a 99.92% detection accuracy on the PKU dataset and outperformed five methods compared therein. Similarly, on the COCO dataset, the proposed method yielded a superior mean average precision than several methods compared therein. Furthermore, for highly challenging conditions in the DAWN dataset, the proposed method was superior in terms of detection accuracy for fog, rain, snow, and sandy conditions.

In the future, the proposed method can be further investigated to detect the occluded vehicles. Moreover, for moving objects, motion blur could also be investigated. Furthermore, a cloud computing-based domain can be introduced to handle the resources of complex machine learning algorithms. Our algorithm could also be investigated for haze images in which there is very limited visibility and thus vehicles are barely visible to human eye. Moreover, the impact of changes in the network structure of each type of a YOLO model could also be further explored on the datasets explored in this study. Finally, our developed method could be integrated with deep learning methods to further explore the research of vehicle detection, tracking, or recognition.

**Author Contributions:** Conceptualization, A.F. and F.H.; methodology, A.F.; software, A.F., K.K., M.S., U.K., Z.M. and F.H.; validation, A.F. and F.H.; formal analysis, A.F., K.K., M.S., U.K., Z.M. and F.H.; investigation, A.F. and F.H.; resources, A.F., K.K., M.S., U.K., Z.M. and F.H.; data curation, A.F.; writing—original draft preparation, A.F., K.K., M.S., U.K., Z.M. and F.H.; writing—review and editing, A.F., K.K., M.S., U.K., Z.M. and F.H.; visualization, A.F. and F.H.; supervision, F.H.; project administration, F.H. All authors have read and agreed to the published version of the manuscript.

**Funding:** This research received no external funding.

**Institutional Review Board Statement:** Not applicable.

**Informed Consent Statement:** Not applicable.

**Data Availability Statement:** Not applicable.

**Conflicts of Interest:** The authors declare no conflict of interest.

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
