# Peer review of "A Fast and Accurate Real-Time Vehicle Detection Method Using Deep Learning for Unconstrained Environments"

_applsci, doi:10.3390/app13053059_

Round 1

Reviewer 1 Report

In the paper, a new dataset containing low density traffic and high density traffic is created, which is used to train YOLOv5. The resulting network is compared to other learning-based approaches on different public datasets.

The authors investigate a challenging and highly-relevant topic. However, the paper has several shortcomings and leaves some open questions:

- The introduction should focus more on the specific topic than on object detection in general.

- The related work section only covers YOLO-based approach but nothing beyond that.

- Algorithm 1 shows the workflow of the paper. However, since this is not an actual algorithm that can be executed, it should rather be visualized using a dataflow diagram or similar instead of using pseudo code.

- Is the labeling tool fully automated or does it just support the manual labeling process?

- The algorithms used for the comparison differs between the three datasets, which makes it impossible to compare them with each other.

- There is no comparison with a YOLOv5-based approach.

- For the COCO and DAWN datasets, there is no comparison with any YOLO-based approach.

- Some of the approaches used for comparison have not been designed for vehicle detection.

- Have all approaches been trained with the same training data?

- The English language and literary style needs to be improved.

Overall, the scientific value of the paper appears to be rather low. The main contribution is creating a new dataset and use it for training an existing model. However, neither the dataset nor the trained model are are made available. Thus, other researches cannot benefit from the finding in this paper.

Reviewer 2 Report

This research article detect and classify vehicles in congested traffic environments shown in publicly available datasets by utilizing the YOLO-v5 architecture.  This paper is well organized and the descriptions have well flowed. To improve the quality of the paper, the authors need to consider the following items for revision.

 Major comments:

1. It is suggested that the author revises English writing.

2. Please check author guidelines for formatting and revision based on journal requirement. (Figure, table, line, spacing, etc).

3. Introduction is well flown, the target need to be clearer based on previous limitations as this work is very common in the file of deep learning work.

4. For data annotation, author used Label Image Tool (LIT) (lines 288-290), and Dark Label Tool (DLT) (Lines 98-299), are they automatic? Is the bounding boxes appear automatically? If yes, then how about the correctness, was it enough, how to define? If no, then please add description about the annotation process. Although, author mentioned data annotation were done manually in abstract, but information in lines 288-290 are seems automatic. Please make a clear description.

5. The methods is well explained. It is needed to explain why author choose YOLO V5? Also need to be more details about the Yolo V5 structure.

6. The results are well flown and explained with figures and tables. Section 4 (lines 369-380) looks like the evaluation procedure and platform, so needs to move to materials and method section, after line 367.

7. The authors are kindly advised to assess the method to compare with several other methods with same data sets (author prepared). The discussion need to be revised with major finding and limitations.

8. If the software or platform (Roboflow, Label Image Tool, Dark Label Image Tool, etc.) are not developed by author, it is recommended to add references.

9. Conclusion and abstract can be revised with major findings, drawbacks and future study.

Minor comments:

1. Is Figure 1 developed by authors? If not reference is needed.

2. Consider the journal table format of table 2-7.

3. Line 378, Average mean Precision (mAP),> Mean Average Precision (mAP)

4. Line 516, remove .(dot) before conclusion.

Author Response

Response is attached in word document.

Reviewer 3 Report

1. Topic is too broad. Various similar papers are available. What is new ?

2. Abstract is vague. need to be concise on the used methods and relevant aspects

3. Introduction has some object detection examples but it is not clear that why we have to start from here. Need clarity on unconstrained environments

4. Only Yolo 4 is discussed in the literature. What are the other aspects. Which other authors are using Yolo 4 for unconstrained environments

5. Propose method requires more clarity. What is the time period for data collection. Statistics needed to be clear

6. Explanation on all figures need to be provided. Images should be provided separately. Box region is not clear in many

7. Discussion need to be detailed.

Author Response

Response is attached in word document.

Round 2

Reviewer 1 Report

In the revised version, the manuscript has significantly been improved. Several sections have been changed, an additional diagram has been added, the related work section has been improved, and further details have been provided. Furthermore, YOLOv5 has been included in the evaluation. There are however a few points that need further improvement:

- While the section on related work now includes conventional methods, it still doesn't cover learning-based approaches that are not based on YOLO. The authors should at least describe all approach that are used in the evaluation. Furthermore, there should be a discussion on their particular shortcomings that are supposed to be addressed by the proposed approaches.

- In the response to the reviewer, the authors say that the algorithms used for comparison have been chosen because their implementations are available and that they have been trained with the same data as the proposed method. Since all approaches have been trained by the authors, it is unclear why different methods are used for the three datasets. Even though the original works have not applied the methods to all datasets, this could be done by retraining.

- Since the authors plan to publish the dataset and the trained model, this information should be added to the manuscript.

Author Response

Response is attached

Reviewer 3 Report

1. As abstract is improved. Suggested title should include deep learning too

2. Introduction is improved

3. Literature has been improved

4. Method has some more points added

5. Simulation results have been improved

6. Discussion is suggested to be detailed. It is just generic. There is one figure too but not referred in the text. 

Author Response

Response is attached
